# Lyapunov exponents and entanglement entropy transition on the noncommutative hyperbolic plane

**Sriram Ganeshan**[1,2]⋆ **and Alexios P. Polychronakos**[1,2]†

**1** Physics Department, City College of the CUNY, New York, NY 10031
**2** ACUNY Graduate Center, New York, NY 10031

⋆ sganeshan@ccny.cuny.edu, † apolychronakos@ccny.cuny.edu

## Abstract

We study quantum dynamics on noncommutative spaces of negative curvature, focusing on the hyperbolic plane with spatial noncommutativity in the presence of a constant magnetic field. We show that the synergy of noncommutativity and the magnetic field tames the exponential divergence of operator growth caused by the negative curvature of the hyperbolic space. Their combined effect results in a first-order transition at a critical value of the magnetic field in which strong quantum effects subdue the exponential divergence for *all* energies, in stark contrast to the commutative case, where for high enough energies operator growth always diverge exponentially. This transition manifests in the entanglement entropy between the 'left' and 'right' Hilbert spaces of spatial degrees of freedom. In particular, the entanglement entropy in the lowest Landau level vanishes beyond the critical point. We further present a non-linear solvable bosonic model that realizes the underlying algebraic structure of the noncommutative hyperbolic plane with a magnetic field.

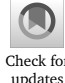
# 1  Introduction

The study of dynamics on negative curvature surfaces has yielded some important results in the subject of classical and quantum chaos [1–6]. Recent developments in the Sachdev-Ye-Kitaev (SYK) model [7–11], a large-$N$ solvable many body quantum system that exhibits "maximal chaos", has shown that the low energy sector in the large-N limit can be described by dynamics on negatively curved spaces. The low energy effective theory corresponding to the SYK model is described by the Schwarzian action, which is invariant under $SL(2,R)$ transformations and possesses a set of conserved charges $J_i$ that generate the $sl(2,R)$ algebra $[J_i, J_k] = \epsilon_{ij}^k J_k$. The corresponding Hamiltonian $H$ is found to be equal to the $SL(2,R)$ Casimir, $H = \frac{1}{2}J^2$, and the energy spectrum and dynamics are thus uniquely determined by the $SL(2,R)$ symmetry. This algebraic structure is closely related to the Landau level problem on the hyperbolic plane which was solved by Comtet and Houston [12, 13]. This connection was identified by Mertens et al [11] who showed that under appropriate limiting conditions the density of states of the Landau level problem on the hyperbolic plane coincides with that of the Schwarzian theory.

It is important to note that the spectrum in the low energy Schwarzian sector of SYK model only accesses the continuous series of the $SL(2,R)$ symmetry that results in exponentially diverging "trajectories"[1] for all energy states [14]. The full Landau level problem, on the other hand, invokes both discrete and continuous series [11, 13, 14]. The discrete series of the $SL(2,R)$ algebra contributes states to the Landau level problem where the magnetic field dominates the negative curvature to yield closed cyclotron orbits.

Our work in this paper is motivated by the question of what variants of the Landau level problem on the hyperbolic plane can subdue the exponential divergence of operator growth in addition to the magnetic field. To this end we consider the quantum dynamics on a noncommutative hyperbolic plane with a constant magnetic field [15].[2]

Noncommutativity in the presence of magnetic field introduces a phase transition as a function of the noncommutativity parameter $\theta$ and magnetic field $B$. For subcritical values $\theta B < 1$, the spectrum consists of both discrete Landau levels and continuous states, similar to the Landau level problem on the hyperbolic plane. However, for $\theta B > 1$ the spectrum consists entirely of Landau levels for all energies with no continuum [15]. Remarkably, this transition can be tracked by studying the non-analyticity in the entanglement entropy between the intrinsic "left" and "right" components of the noncommutative Hilbert space. Quantifying this transition using entanglement entropy is the main result of this work. We also derive a solvable non-linear bosonic Hamiltonian that realizes underlying algebraic structure of the non-commutative hyperbolic plane with a magnetic field. The advantage of this model is that

---

[1]For quantum systems trajectories does not have any physical meaning beyond the semiclassical limit. We consider operator spreading rate as a proxy for the classical Lyapunov exponent. The operator spreading rate and the Lyapunov exponent are closely related for the dynamics on the hyperplane.

[2]We point out that the noncommutativity in the lowest Landau level arises from the projection operation into the degenerate ground state manifold and consequently the coordinates become equivalent to the momentum. This projection makes the momentum non-dynamical. In noncummutative spaces, we have *independent* coordinates with a deformed commutation relation between the coordinates via the non-commutativity parameter $\theta$ as well as the usual non-commutativity of the momenta via the magnetic field $B$. Consequently, we have *two* length scales in the problem given by $1/\sqrt{B}$ and $\sqrt{\theta}$.

the curvature, noncommutativity parameter and the magnetic field enters as parameters of the Hamiltonian.

This paper is organized as follows. We begin by calculating Lyapunov exponents across the spectral transition on the noncommutative hyperbolic space with a magnetic field (Sec 2). In Sec 3, we introduce the notion of entanglement between the "left" and "right" Hilbert spaces of spatial degrees of freedom on the noncommutative manifold and calculate entanglement entropy (EE) across the spectral transition for NC plane and hyperplane. In Sec. 4, we derive a non-linear solvable bosonic Hamiltonian that realizes the algebraic structure of the noncommutative hyperbolic plane with magnetic field . Finally, we address the role of spatial compactness on the entanglement entropy transition by considering the case of non-commutative sphere (Sec. 5). We end the paper with conclusions and future directions in Sec. 6.

## 2 Operator spreading on the noncommutative hyperbolic plane

In this section we study the quantum dynamics of operators on the noncommutative hyperbolic plane. For brevity we will refer from now on to the noncommutative sphere, noncommutative plane and noncommutative hyperbolic plane as NC plane, NC sphere and NC hyperplane.

The full dynamics are in principle determined by the energy spectrum. The spectrum of a charged particle on the NC hyperplane, which includes Landau levels and a continuum, was derived in Ref. [15] and is reviewed in the entanglement section (Sec 3). Its flow is presented in Fig. 2. Below a critical value of the magnetic field $B < 1/\theta$ the energy spectrum consists of both discrete and continuous parts. The continuum arises for energies above a certain threshold for which the cyclotron orbits are not closed due to the negative curvature of the hyperbolic ($AdS_2$) manifold. However, for $B > 1/\theta$, the continuous part of the spectrum is completely eliminated. From this spectrum we could in principle determine the quantum evolution of operators as a function of energy, magnetic field and the noncommutativity parameter. In the sequel, however, we will work purely with operator relations to find their evolution without any explicit reference to the spectrum.

As is standard in noncommutative theories, the momentum operator can be realized via a second copy of space operators. The two copies can be though of as arising from the left and right actions of coordinate operators and should not be misconstrued as enlarging the Hilbert space. (A a quick review of noncommutative quantum mechanics is given in the appendix). The two components of the algebra (the "space" part $R$ and "covariant" part $K$) mutually commute and satisfy the $SL(2,R)$ algebra

$$[R_j, R_k] = i \, \epsilon_{jk}{}^l R_l, [K_j, K_k] = i \, \epsilon_{jk}{}^l K_l \, , \quad [R_j, K_k] = 0 \, . \tag{1}$$

Indices are raised and lowered with the Minkowski metric ($\eta_{ij}$, $\eta_{11} = \eta_{22} = -\eta_{33} = 1$, otherwise $\eta_{jk} = 0$). Space coordinates are represented as $X_i = (\theta/a)R_i$, with $\theta$ the noncommutativity parameter and $a$ the radius of the hyperbolic plane (that is, $-1/a^2$ is the curvature). The square of the space component $-a^2 = X^2 = (\theta^2/a^2)R^2$, with

$$R^2 = R_i R^i = R_1^2 + R_2^2 - R_3^2 = r(1-r) \tag{2}$$

must be negative, corresponding to a negative curvature Euclidean signature hyperbolic plane, and therefore the Casimir $R^2$ must be negative. (A positive Casimir would give, instead, a $(1+1)$-dimensional anti-de Sitter spacetime.) The noncommutative space parameter $\theta$ is given by the relation

$$\theta = \frac{a^2}{\sqrt{r(r-1)}} \quad \Rightarrow \quad [X_1, X_2] = i \frac{\theta}{a} X_3 \, . \tag{3}$$

The Casimir of the covariant part, on the other hand,

$$K^2 = K_i K^i = k(1-k) \,, \quad k = r + b \tag{4}$$

is, in principle, not constrained. The difference $b = k - r$ quantifies the magnetic field through the relation (see Appendix section 7 for details)

$$(1 - \theta B)^2 = \frac{r(r-1)}{k(k-1)} \,. \tag{5}$$

Note that neither $r$ nor $b$ need be quantized and there is no monopole quantization. The commutative limit is recovered as $r \to \infty$ with $a$ fixed, and in that limit $b = Ba^2$.

The generators of the NC hyperplane translations and rotations are $J_i = R_i + K_i$, with $J^2 = J_1^2 + J_2^2 - J_3^2$ and $[J_i, R_j] = i\epsilon_{ij}{}^k R_k$. The Hamiltonian for the NC hyperbolic plane in the presence of magnetic field is

$$H = \frac{\gamma}{2a^2} J^2 + \frac{B^2 a^2}{2\gamma} \,, \quad \gamma = 1 - \theta B \,. \tag{6}$$

In the following, we will work with the rescaled coordinate $R_i$ instead of $X_i$ and will also rescale time $t \to (a^2/\gamma)t$, therefore rescaling the Hamiltonian to $\frac{1}{2}J^2$ plus a constant, to shed all superfluous factors.

The Heisenberg evolution $\dot{R}_i = -i[R_i, H]$ of the rescaled coordinate $R_i$ can be written as

$$\dot{R}_i = M_i^j R_j \,, \quad M_i^j = \epsilon_i{}^{kj} J_k + i\delta_i^j. \tag{7}$$

The operator-matrix $M$ satisfies the following useful properties that will allow us to simplify the integrated equation of motion,

$$M^2 = iM + J^2 P \,, \quad P_i^j = \delta_i^j - \frac{1}{J^2} J_i J^j, \quad MP = PM = M \,, \quad P^2 = P \,, \quad [P, J^2] = 0.$$

The equation of motion for $R_i$ can then be integrated to

$$R_i(t) = \left(e^{Mt}\right)_i^j R_j(0), \quad e^{Mt} = \sum_{n=0}^{\infty} \frac{t^n}{n!} M^n \,. \tag{8}$$

We now use the properties of the $M$ matrix to evaluate the exponential $\exp Mt$ in terms of new operators $A = M + i\alpha P \,, \ B = M + i\beta P$ for some real $\alpha, \beta$. The operators $A, B$ will then satisfy

$$A^2 = i(1+2\alpha)A, \ B^2 = i(1+2\beta)B, \ AB = BA = i(1+\alpha+\beta)M + (J^2 - \alpha\beta)P \,. \tag{9}$$

Choosing $\alpha$ and $\beta$ to be the roots of the quadratic equation $s^2 + s + J^2 = 0$ we can make $AB = BA = 0$. From the definition of $A$ and $B$ we obtain $M = c_a A + c_b B$, with $c_a = -\beta/(\alpha - \beta)$ and $c_b = \alpha/(\alpha - \beta)$. Putting all these properties together we get

$$M^n = c_a^n (i(1+2\alpha))^{n-1} A + c_b^n (i(1+2\beta))^{n-1} B \,, \quad n \geq 1 \tag{10}$$

$$\text{and thus} \quad e^{Mt} = 1 + \frac{e^{ic_a(1+2\alpha)t} - 1}{i(1+2\alpha)} A + \frac{e^{ic_b(1+2\beta)t} - 1}{i(1+2\beta)} B \,. \tag{11}$$

Finally, substituting $\alpha = \frac{-1+\sqrt{1-4J^2}}{2}, \beta = \frac{-1-\sqrt{1-4J^2}}{2}$ we obtain the full time dependent operator growth as,

$$R(t) = \left(e^{it/2}\left[\frac{\sin\omega_0 t}{\omega_0}(M - iP/2) + \cos\omega_0 t \, P\right] + 1 - P\right) R(0)$$

$$\text{with} \quad \omega_0 = \sqrt{-J^2 + 1/4}. \tag{12}$$

The above expression completely quantifies how the operator vector $R(t)$ evolves or spreads in time $t$.

There are two distinct regimes in the operator evolution depending on whether $\omega_0$ is real or complex. $\omega_0^2$ can take either sign due to the $SL(2,R)$ nature of the Casimir $J^2 = J_1^2 + J_2^2 - J_3^2 = j(1-j)$. We observe that:

*a.* For $j$ real, $J^2 \leq 1/4$, so $\omega_0^2 \geq 0$ and $R(t)$ is oscillatory. Real $j$ yields the discrete representations of $SL(2,R)$ which correspond to discrete Landau levels. This is the case where orbits close as the magnetic field dominates curvature.

*b.* For $j = \frac{1}{2} + is$, with $s$ real, $J^2 \geq 1/4$, so $\omega_0^2 < 0$ and $R(t)$ spreads exponentially. In this case $\omega_0$ is nothing but the energy dependent *Lyapunov exponent*. Such Casimirs yield continuous representations of $SL(2,R)$ which correspond to the continuous part of the energy spectrum. This is the case where the orbits do not close due to the negative curvature dominating the magnetic field term.

Since $J_i = R_i + K_i$ the representations of $J$ are determined by those of $R$ and $K$. From the representation theory of $SL(2,R)$ it follows that in the subcritical case $B < 1/\theta$ there are $n = [b-1/2]$ discrete Landau levels above which lies a continuum, while in the overcritical cased $B > 1/\theta$ there is only an infinite number of Landau levels (these facts will be shown explicitly in the section of entropy calculation). So a corollary of the above evolution is that there is *no* exponential spreading of the operator $R(t)$ in the overcritical case for any energy, since the continuous spectrum is absent. Remarkably, the operator $R(t)$ is prevented from spreading exponentially for all energies despite being defined on a negative curvature surface.

## 3   Entanglement entropy transition

The Hilbert space structure of the noncommutative space allows us to quantify the spectral transition discussed in the previous section through non-analyticity in the entanglement entropy at the critical point. We now proceed to demonstrate that the transition from exponential spreading to oscillatory behavior of operators is quantified by the entanglement entropy between the two Hilbert space components of the noncommutative space. We are mainly interested in the NC hyperplane in a constant magnetic field, but we will first examine the instructive (and simpler) case of the NC plane.

### 3.1   Entanglement transition on the noncommutative plane

The NC plane with noncommutativity parameter $\theta$ and magnetic field $B$ is represented by two Heisenberg algebras or, equivalently, by two harmonic oscillator algebras. One realization of the NC coordinates $x_1, x_2$ and their canonical momenta $p_1, p_2$ is [16]

$$
\begin{aligned}
x_1 &= \sqrt{\theta}\,\alpha_1 & p_1 &= \frac{1}{\sqrt{\theta}}(\beta_1 + \sqrt{|\gamma|}\,\alpha_2) \\
x_2 &= \sqrt{\theta}\,\beta_1 & p_2 &= \frac{1}{\sqrt{\theta}}(-\alpha_1 - \mathrm{sgn}(\gamma)\sqrt{|\gamma|}\,\beta_2),
\end{aligned}
\tag{13}
$$

with

$$
[\alpha_i, \beta_j] = i\delta_{ij}\,, \quad [\alpha_i, \alpha_j] = [\beta_i, \beta_j] = 0\,, \quad \gamma = 1 - \theta B\,,
\tag{14}
$$

such that

$$
[x_1, x_2] = i\theta\,, \quad [p_1, p_2] = iB\,, \quad [x_i, p_j] = i\delta_{ij}\,.
\tag{15}
$$

We call $\alpha_1, \beta_1$ the "left" coordinates and $\alpha_2, \beta_2$ the "right" ones (a terminology inherited from the operator representation of NC fields). The Hamiltonian of a charged particle on the NC

plane, which can also include a harmonic oscillator potential, reads

$$H = \frac{1}{2}\left[p_1^2 + p_2^2 + \omega^2(x_1^2 + x_2^2)\right].$$ (16)

It is a quadratic expression on the phase space, and an appropriate Bogolyubov transformation can always diagonalize it and express it in terms of a new pair of decoupled harmonic oscillators $A_i, A_i^\dagger$. The appearance of $\text{sgn}(\gamma)$ in (13) signals that the situations $B > \theta^{-1}$ ($\gamma > 0$) and $B < \theta^{-1}$ ($\gamma < 0$) are qualitatively different and have to be treated separately. $B = \theta^{-1}$ is a critical value of the magnetic field and separates a 'subcritical' from an 'overcritical' phase.

In the subcritical phase $B < \theta^{-1}$ the appropriate diagonalizing Bogolyubov transformation is

$$A_1 = \cosh\lambda\, a_1 - i\sinh\lambda\, a_2^\dagger, \quad A_2 = \cosh\lambda\, a_2 - i\sinh\lambda\, a_1^\dagger,$$ (17)

with

$$a_i = \frac{1}{\sqrt{2}}(\alpha_i + i\beta_i), \quad \tanh 2\lambda = -\frac{2\sqrt{\gamma}}{1 + \gamma + \omega^2\theta^2}$$ (18)

and turns the Hamiltonian into

$$H = \omega_+ A_1^\dagger A_1 + \omega_- A_2^\dagger A_2 + \frac{\omega_+ + \omega_-}{2},$$ (19)

with

$$\omega_\pm = \frac{1}{2}\left|\sqrt{(\omega^2\theta - B)^2 + 4\omega^2} \pm (\omega^2\theta + B)\right|.$$ (20)

We shall calculate the entanglement between the "left" and "right" spaces for the ground state $|\Omega\rangle$, that is, the state satisfying the property

$$A_1|\Omega\rangle = A_2|\Omega\rangle = 0.$$ (21)

The oscillators $a_1, a_1^\dagger$ and $a_2, a_2^\dagger$ act on the left and right Fock spaces, respectively. The full Hilbert space is spanned by Fock states $|n_1, n_2\rangle$. From the relation

$$a_1^\dagger a_1 - a_2^\dagger a_2 = A_1^\dagger A_1 - A_2^\dagger A_2,$$ (22)

which is a corollary of (17), we deduce that $(a_1^\dagger a_1 - a_2^\dagger a_2)|\Omega\rangle = 0$ and thus $|\Omega\rangle$ must have the form

$$|\Omega\rangle = \sum_{k=0}^\infty C_k |k, k\rangle.$$ (23)

Implementing the ground state condition (21) yields the relation $C_k = i\tanh\lambda\, C_{k-1}$ which, upon normalizing $\langle\Omega|\Omega\rangle = 1$ leads to the expression $C_k = \cosh^{-1}\lambda\, (i\tanh\lambda)^k$. The state $|\Omega\rangle$ corresponds to the pure density matrix $\rho = |\Omega\rangle\langle\Omega|$. Tracing it over the right space gives the mixed state on the left space

$$\rho_L = \text{tr}_R \rho = \sum_{k=0}^\infty |C_k|^2 |k\rangle\langle k|.$$ (24)

The von Neumann entropy of this state is the entanglement entropy between the spaces for the ground state which after using the expression for $C_n$ yields

$$S_0 = -\sum_{k=0}^\infty |C_k|^2 \ln|C_k|^2 = \cosh^2\lambda \ln\cosh^2\lambda - \sinh^2\lambda \ln\sinh^2\lambda.$$ (25)

The entanglement entropy goes to zero as $B$ approaches the critical value $\theta^{-1}$ and has a maximum at $B = -\omega^2\theta$, dropping back to zero as $B \to -\infty$. In the pure magnetic case $\omega = 0$ the entanglement entropy diverges at $B = 0$.

In the overcritical phase $B > \theta^{-1}$ the Hamiltonian retains its form (19) with frequencies $\omega_\pm$ as in (20). The diagonalizing transformation in this case, however, becomes

$$A_1 = \cos\lambda\, a_1 + i\sin\lambda\, a_2\,, \quad A_2 = \cos\lambda\, a_2 + i\sin\lambda\, a_1\,, \tag{26}$$

with

$$\tan 2\lambda = \frac{2\sqrt{-\gamma}}{1 + \gamma - \omega^2\theta^2}\,. \tag{27}$$

It is not a proper Bogolyubov transformation, as it does not mix creation and annihilation operators. The ground state $A_1\,|\Omega\rangle = A_2\,|\Omega\rangle = 0$ is now annihilated by $a_1$ and $a_2$, so it is the unentangled vacuum state $|0,0\rangle$. Thus the entanglement entropy for the ground state remains zero throughout the overcritical domain.

It is of interest to examine the entanglement property of Landau states in the case $\omega = 0$, as this will be the situation most relevant to the NC sphere and hyperplane. The ground state $|\Omega\rangle$ in that case goes over to the minimum angular momentum (minimum cyclotron radius) state in the lowest Landau level (LLL). Its entanglement entropy in the subcritical phase takes the explicit form

$$S_0 = -\ln|1 - \gamma| - \frac{\gamma}{1 - \gamma}\ln\gamma = -\ln|\theta B| - \left[(\theta B)^{-1} - 1\right]\ln(1 - \theta B)\,. \tag{28}$$

The behavior of $S$ as a function of $B$ is shown in Figure 2. The entropy is infinite for $B = 0$, then decreases to zero at the critical point $B = \theta^{-1}$ and remains vanishing into the overcritical region. For large negative $B$ it falls off like $\sim \ln|\theta B|/|\theta B|$.

The entanglement entropy will remain the same for any other minimal radius state in the LLL, created by a translation of $|\Omega\rangle$ on the NC plane. Such translations are generated by the magnetic translation generators

$$D_1 = \frac{1}{\theta B}(p_1 - Bx_2)\,, \quad D_2 = \frac{1}{\theta B}(p_2 + Bx_1)\,. \tag{29}$$

$D_1$ and $D_2$ are sums of left and right operators, and therefore the unitary translations they generate will be products of unitary left and right operators. Any such factorized unitary transformation of the state $|\Omega\rangle$ preserves the entanglement entropy.

The same is not true, however, for higher angular momentum (non-minimal radius) states on the LLL, or for states in higher Landau levels. To examine the entropy of such states we focus on excited states of one of the oscillators, say $A_1$. Since the entanglement entropy is symmetric under exchange of $A_1$ and $A_2$, such excitations probe the entropy of both higher angular momentum LLL states and minimal radius higher Landau level states. Such an $n$th excited state, denoted $|\Omega_n\rangle$, satisfies

$$A_1^\dagger A_1\,|\Omega_n\rangle = n\,|\Omega_n\rangle\,, \quad A_2\,|\Omega_n\rangle = 0\,. \tag{30}$$

In the subcritical phase, relation (22) again implies that the state must be of the form

$$|\Omega_n\rangle = \sum_{k=0}^{\infty} C_{n,k}\,|k + n, k\rangle \tag{31}$$

and on this state it is enough to implement the condition $A_2\,|\Omega_n\rangle = 0$, the other condition in (30) being automatically satisfied. This implies the relation

$$C_{n,k} = i\tanh\lambda\,\sqrt{\frac{n+k}{k}}\,C_{n,k-1}\,, \tag{32}$$

from which we obtain

$$C_{n,k} = C_{n,0} \, (i \tanh\lambda)^k \sqrt{\frac{(n+k)!}{n! \, k!}} \,. \tag{33}$$

The overall constant $C_{n,0}$ is fixed by the normalization condition $\sum_n |C_{k,n}|^2 = 1$ and we find

$$C_{n,k} = \cosh^{-n-1}\lambda \, (i \tanh\lambda)^k \sqrt{\frac{(n+k)!}{n! \, k!}} \,. \tag{34}$$

The entanglement entropy is again given by the standard formula

$$S_n = -\sum_{k=0}^{\infty} |C_{n,k}|^2 \ln |C_{n,k}|^2 \,. \tag{35}$$

In the overcritical phase, relation (22) does not hold; instead, we have

$$a_1^\dagger a_1 + a_2^\dagger a_2 = A_1^\dagger A_1 + A_2^\dagger A_2 \,. \tag{36}$$

Therefore, the state $|\Omega_n\rangle$ is of the form

$$|\Omega_n\rangle = \sum_{k=0}^{n} C_{n,k} \, |k, n-k\rangle \,, \quad A_2 \, |\Omega_n\rangle = 0 \,. \tag{37}$$

Solving for $C_{n,k}$ and implementing the normalization condition yields

$$C_{n,k} = \cos^n\lambda \, (-i \tan\lambda)^k \sqrt{\frac{n!}{k!(n-k)!}} \tag{38}$$

and the entropy is given again by (35) with the sum truncated to $n+1$ terms. It has a maximum value of $S_{n,max} = \ln(n+1)$ achieved for $\tan\lambda = 1$. $S_n$ increases with increasing $n$ in both the subcritical and overcritical phases.

For the case of a pure magnetic field ($\omega = 0$) the entanglement entropy becomes a function of $\gamma$ and the expressions for $S_k(\gamma)$ simplify, although they do not become elementary functions. We point out that the entanglement entropy, for all $n$ and for both phases, obeys the duality relation

$$S_n(\gamma) = S_n(\gamma^{-1}) \,, \tag{39}$$

with its values at the two self-dual points $\gamma = \pm 1$ being

$$S_n(\gamma = 1) = S_n(B = 0) = \infty \,, \quad S_n(\gamma = -1) = S_n(B = 2\theta^{-1}) = \ln(n+1) \,, \tag{40}$$

while the critical ($\gamma = 0$) and infinite ($\gamma = \pm\infty$) magnetic fields are dual to each other, with vanishing entropy:

$$S_n(B = \theta^{-1}) = S_n(B = \pm\infty) = 0 \,. \tag{41}$$

The duality $S(\gamma) = S(\gamma^{-1})$ for $\omega = 0$ can be understood as a mapping between the "left" and "right" Hilbert spaces with respect to which the the entanglement entropy is calculated. From Eq. (13) in the main text, consider the momenta $p_{1,2}$ with $\gamma > 0$,

$$p_1 = \frac{1}{\sqrt{\theta}} (b_1 + \sqrt{\gamma}\alpha_2) \,, \quad p_2 = \frac{1}{\sqrt{\theta}} (-\alpha_1 - \sqrt{\gamma}b_2) \,. \tag{42}$$

We now define new operators $(\tilde{\alpha}_1, \tilde{b}_1) = (b_1, -\alpha_1)$ and $(\tilde{\alpha}_2, \tilde{b}_2) = (-b_2, \alpha_2)$, in terms of which the momenta $p_{1,2}$ can be written as,

$$\frac{1}{\sqrt{\gamma}} p_1 = \frac{1}{\sqrt{\theta}} \left( \tilde{b}_2 + \frac{1}{\sqrt{\gamma}} \tilde{\alpha}_1 \right) \quad -\frac{1}{\sqrt{\gamma}} p_2 = \frac{1}{\sqrt{\theta}} \left( -\tilde{\alpha}_2 - \frac{1}{\sqrt{\gamma}} \tilde{b}_1 \right) \,. \tag{43}$$

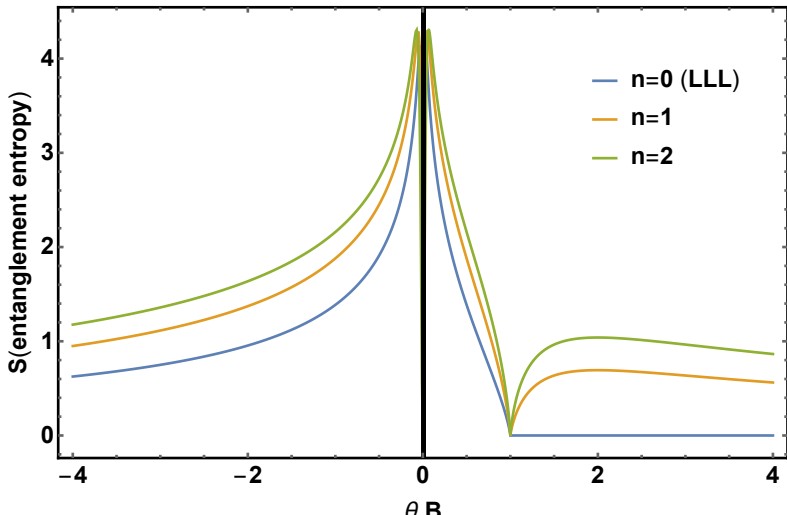

Figure 1: Entanglement entropy for the minimum angular momentum states corresponding to lowest, first and second Landau levels.

Thus a rescaling and a parity transform $p_2 \to -p_2$ of the momentum, and a mapping of spaces $1 \leftrightarrow 2$ (with some canonical transformation of coordinates in each) amount to $\gamma \to 1/\gamma$. Since the Hamiltonian $H = \frac{1}{2}(p_1^2 + p_2^2)$ maps to $1/\gamma H$ under the above transformation, the ground state remains unchanged and has same entanglement entropy. Note that the coordinates $x_{1,2}$ map to themselves under the same transformation. Thus for $\omega \neq 0$ the Hamiltonian $H = \frac{1}{2}(p_1^2 + p_2^2) + \frac{\omega^2}{2}(x_1^2 + x_2^2)$ does not map to a scaled version of itself and thus breaks the duality. Another way to express the duality is by using the "Seiberg-Witten" mapped magnetic field $\tilde{B} = B/(1 - \theta B)$, which leads to $S(B) = S(-\tilde{B})$. Figure 1 shows the behavior of the entanglement entropy of the lowest angular momentum $n = 0$, $n = 1$ and $n = 2$ Landau level states. $S_0 \leq S_1 \leq S_2$ and they all vanish at the critical point, but $S_1$ and $S_2$ become positive in the overcritical region while $S_0$ remains zero.

An important remark is in order: the calculation for the entanglement entropy and the resulting transition sensitively depends on our choice of "left" $(\alpha_1, \beta_1)$ and "right" $(\alpha_2, \beta_2)$ Hilbert spaces. A different factorization of the full Hilbert into two components would have revealed no non-analytic behavior of the entanglement entropy near $\theta B = 1$. E.g., choosing $C_1 = p_1 + ip_2$ and $C_2 = p_2 + Bx_1 + i(p_1 - Bx_2)$ we have

$$[C_1, C_1^\dagger] = 2B, \quad [C_2, C_2^\dagger] = 2\gamma B, \quad [C_1, C_2] = [C_1, C_2^\dagger] = 0. \tag{44}$$

So $C_1, C_1^\dagger$ and $C_2, C_2^\dagger$ are two decoupled oscillators with their own Fock space, and the Hamiltonian (for $\omega = 0$) is $H = \frac{1}{2}C_1^\dagger C_1$. So we can always choose the energy eigenstates to be factorized states in the two oscillators with zero entanglement. Note, however, that such states would not be minimum space uncertainty (cyclotron radius) states, unlike the states we considered in our calculation. Note, further, that the second oscillator would become degenerate at the critical point $\gamma = 0$ and would invert creation and annihilation in the supercritical phase.

This freedom is absent for NC spaces with curvature (hyperplane and sphere considered here), where the factorization into left and right $SL(2, R)$ or $SU(2)$ algebras is unique. The non-analyticity in the entanglement entropy will have physical consequences due to the uniqueness of the partition between the left and right spaces. We will calculate this entanglement entropy in the following sections.

## 3.2 NC hyperplane

The calculation of the entanglement entropy on the NC hyperplane proceeds along similar lines as the one for the NC plane, with some important differences due to the curved nature of the space.

The basic algebraic elements for the NC hyperplane have been laid out in section 2 and to facilitate the reader we repeat here the relevant facts. The space ("left") part $R$ and covariant ("right") part $K$ satisfy the $SL(2,R)$ algebra

$$[R_j, R_k] = i\,\epsilon_{jk}{}^l R_l\,, \quad [K_j, K_k] = i\,\epsilon_{jk}{}^l K_l \tag{45}$$

and have Casimirs

$$R^2 = R_1^2 + R_2^2 - R_3^2 = r(1-r) < 0\,, \quad K^2 = K_i K^i = k(1-k)\,, \quad k = r+b\,. \tag{46}$$

The generators of the hyperplane translations and rotations are

$$J_i = R_i + K_i \tag{47}$$

in analogy with the angular momentum on the sphere.

For each value of $r$ (or $k$) there are actually two irreducible representations, denoted $D_r^\pm$, differing by the sign of $R_3$. Specifically

$$R_3\,|r, m\rangle_\pm = m\,|r, m\rangle_\pm\,, \quad m = \pm r, \pm(r+1), \pm(r+2), \dots \tag{48}$$

That is, for the $+$ $(-)$ irrep the eigenvalues of $R_3$ are above $r$ (below $-r$) respectively. So $D_r^+$ has a lowest weight state $|r, r\rangle_+$ while $D_r^-$ has a highest weight state $|r, -r\rangle_-$. In the sequel we will eliminate the subscripts $\pm$ in states as the sign of $R_3 = m$ fully determines their nature. The action of $R_+$ and $R_-$ on states is

$$R_\pm\,|r, m\rangle = \sqrt{(r \pm m)(-r \pm m + 1)}\,|r, m \pm 1\rangle\,. \tag{49}$$

These representations are relevant to the subcritical and overcritical case, which are qualitatively different.

### 3.2.1 Subcritical case

The NC hyperplane with a magnetic field in the subcritical domain is reproduced by the representation $D_r^- \otimes D_k^+$. The choice of signs is related to the sign of $\theta$, the opposite sign choice $D_r^+ \otimes D_k^-$ corresponding to a parity transformation. (For full details we refer the reader to [15].) The difference $k - r = b$ is related to the magnetic field per unit curvature area, as also stated in section 2. Specifically, the noncommutativity parameter $\theta$ and the magnetic field $B$ are given in terms of the space curvature $1/a$ and the Casimirs $r$ and $k = r + b$ by the relations

$$\theta = \frac{a^2}{\sqrt{r(r-1)}}\,, \quad \gamma = 1 - \theta B = \sqrt{\frac{r(r-1)}{k(k-1)}}\,. \tag{50}$$

In the planar limit $a \to \infty$, $a^2/r \to \theta$, while in the commutative limit $r \to \infty$, $b = Ba^2$. The Hamiltonian is essentially the quadratic Casimir

$$H = \frac{\gamma}{2a^2}J^2 + \frac{B^2 a^2}{2\gamma} \tag{51}$$

and its eigenvalues and eigenstates are given by the quadratic Casimir of irreps of $J$.

The decomposition of $D_r^- \otimes D_k^+$ into irreps of $J$ contains, in general, both discrete and continuous representations. The discrete ones constitute Landau levels, while above a certain energy the spectrum becomes a continuum. We are interested in the Landau levels, so we consider only the discrete irreps.

Landau level $n$ consists of normalizable states in a $D_{b-n}^+$ irrep of $J$ (assuming $b > 0$, so that $k > r$; otherwise $D_{-b-n}^-$). Focusing on the state $|b-n, b-n\rangle$ of lowest $J_3$ as a representative in Landau level $n$, this state will be given by a superposition of $D_r^- \otimes D_k^+$ states

$$|b-n, b-n\rangle = \sum_{k=0}^{\infty} D_k \, |r, -r-n-k; r+b, r+b+k\rangle \,, \tag{52}$$

with $D_k$ Clebsch-Gordan-type coefficients and $n \geq 0$. (States with $J_3 > b$ cannot be bottom states.) It satisfies

$$J_- |b-n, b-n\rangle = 0 \,. \tag{53}$$

Using (49) the above condition yields

$$D_{k+1} = -D_k \sqrt{\frac{(n+k+1)(2r+n+k)}{(k+1)(2r+2b+k)}} \,, \tag{54}$$

from which we obtain

$$D_k = C(-1)^k \sqrt{\frac{(n+k)! \, \Gamma(2r+n+k)}{k! \, \Gamma(2r+2b+k)}} \,, \tag{55}$$

with $C$ an overall constant.

Up to now there was no restriction on $n$. Since this is a $D^+$ representation, however, $J_3 \geq 0$ and thus $n \leq b$. The full allowed values of $n$ are found by imposing the normalizability condition on $|b-n, b-n\rangle$:

$$\sum_{k=0}^{\infty} |D_k|^2 < \infty \quad \Rightarrow \quad \sum_{k=0}^{\infty} \frac{(n+k)! \, \Gamma(2r+n+k)}{k! \, \Gamma(2r+2b+k)} < \infty \,. \tag{56}$$

For large $k$ the terms in the above series behave as $\sim k^{-2(b-n)}$ and summability requires

$$-2(b-n) < -1 \quad \text{or} \quad n < b - \tfrac{1}{2} \,. \tag{57}$$

Altogether we have

$$n = 0, 1, \ldots, n_b \quad \text{for} \quad b = n_b + \alpha \,, \quad \tfrac{1}{2} < \alpha \leq \tfrac{3}{2} \,. \tag{58}$$

We recover the (finite) number of Landau levels that exist for a subcritical magnetic field. All states in Landau level $n$ have a Casimir $J^2 = (b-n)(1-b+n) < \frac{1}{4}$ and energy

$$E_n = \frac{\gamma}{2a^2} \left[ \tfrac{1}{4} - \left(b - \tfrac{1}{2} - n\right)^2 \right] + \frac{B^2 a^2}{2\gamma} \,. \tag{59}$$

The energy increases with $n$ since $n < b - \frac{1}{2}$. The continuum of (scattering-normalized) states starts at a positive Casimir $J^2 = \frac{1}{4}$ and threshold energy

$$E_t = \frac{\gamma}{8a^2} + \frac{B^2 a^2}{2\gamma} \,. \tag{60}$$

States in the continuum have $J^2 > \frac{1}{4}$ and $J_3 = b + n$, $n = 0, \pm 1, \pm 2, \ldots$ and do not satisfy a condition $J_- |\psi\rangle = 0$. When $b = n_b + \frac{1}{2}$ ($\alpha = \frac{1}{2}$) a new Landau level "peels off" the continuum (see Fig. 2).

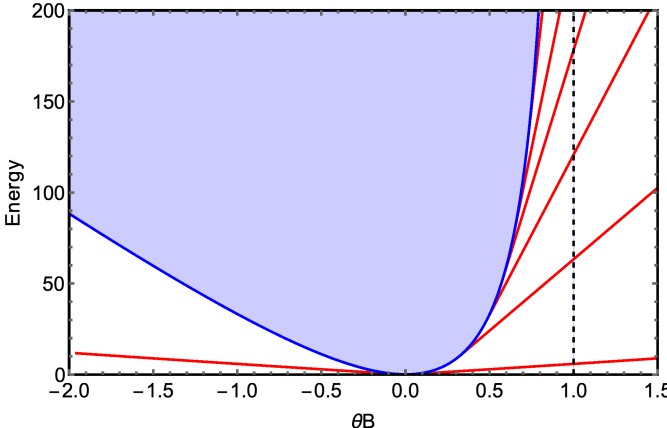

Figure 2: Energy spectrum as a function of magnetic field for the noncommutative hyperbolic plane. Blue shaded part corresponds to the continuum states. Red lines are Landau levels and the vertical black dashed line corresponds to $\theta B = 1$.

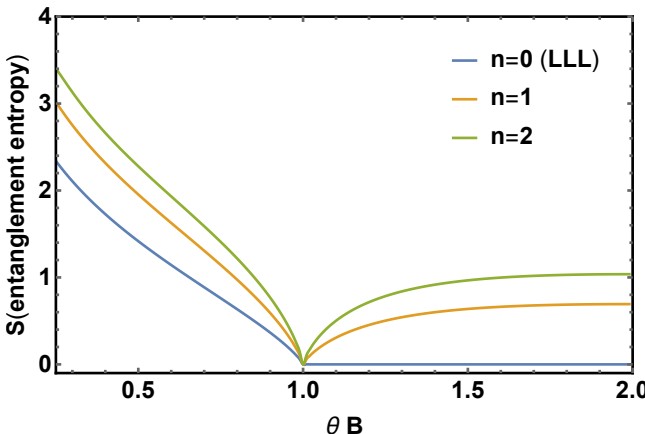

Figure 3: Entanglement entropy as a function of magnetic field for the noncommutative hyperbolic plane (only Landau Level states are considered).

Evaluating the normalization sum in (56) and setting it to 1 yields

$$C^{-2} = \frac{n!\,\Gamma(2r+n)!\,\Gamma(2b-2n-1)}{\Gamma(2r+2b-n-1)\,\Gamma(2b-n)}, \tag{61}$$

which fixes the density matrix probabilities between $R$ and $K$ as

$$|D_k|^2 = \frac{(n+k)!\,\Gamma(2r+2b-n-1)\,\Gamma(2b-n)\,\Gamma(2r+n+k)}{n!\,k!\,\Gamma(2r+n)\,\Gamma(2b-2n-1)\,\Gamma(2r+2b+k)}. \tag{62}$$

The entanglement entropy is given by the standard expression

$$S_n = -\sum_{k=0}^{\infty} |D_k|^2 \ln |D_k|^2. \tag{63}$$

The ground state corresponds to the lowest Landau level $n = 0$, which exists for $b > \frac{1}{2}$, and has the smallest entanglement entropy among Landau states. Entropy increases monotonically for higher levels. For a given Landau level the entropy is infinite as it comes to existence by peeling off the continuum and decreases monotonically with increasing magnetic field (increasing $b$). See Fig. 3 for entanglement entropy for the lowest three LL's across the spectral

transition. Figure 3 shows entanglement entropy for the lowest three LLs before merging into the continuum.

Negative magnetic field corresponds to $k < r$ or $b < 0$. In principle we have to consider the discrete $J$-irreps $D^-_{-b-n}$ in that case. Alternatively, a parity transformation turns the representation of the NC hyperplane to $D^+_r \times D^-_{r-b}$, and the results can be obtained by the formula for $b > 0$ by exchanging $r$ and $k$; that is, by taking $b \to |b|$ and $r \to r - |b|$ in formula (62). We obtain

$$|D_k|^2 = \frac{(n+k)!\,\Gamma(2r-n-1)\,\Gamma(2|b|-n)\,\Gamma(2r-2|b|+n+k)}{n!\,k!\,\Gamma(2r-2|b|+n)\,\Gamma(2|b|-2n-1)\,\Gamma(2r+k)}\,. \tag{64}$$

Since $k = r + b$ cannot be negative, $|b|$ has an upper bound at $r$. Therefore there are a finite number of Landau levels that can form for negative $B$, which is a purely noncommutative effect. The limit $B \to -\infty$ is achieved as $|b| \to r$. Note that the entanglement entropy for the LLL state $n = 0$ vanishes as $B \to -\infty$ but the entanglement entropy of higher Landau level states does not vanish, unlike the NC plane.

As the magnetic field increases it reaches the critical value $B = 1/\theta$ for $b \to \infty$. So an infinity of Landau levels forms until the critical point is reached. In the $b \to \infty$ limit $D_0 \to 1$ and $D_k \to 0$ for all $n$ and $k > 0$, and thus all Landau level states have vanishing entanglement entropy.

### 3.2.2 Overcritical case

The situation changes qualitatively for $B > 1/\theta$. $b$ is finite and decreasing with increasing $B$, but $\gamma$ becomes negative. The realization of the NC hyperplane requires the $R \times K$ representation $D^-_r \times D^-_k$ and $J$ now contains only discrete irreps and no continuum. The infinite number of Landau states that form as $B \to 1/\theta$ persist into the overcritical region with increasing energy. Note that there is no discontinuity as we transit from $D^-_r \times D^+_k$ to $D^-_r \times D^-_k$ at $b = \infty$ since all higher states of $D^\pm_k$ decouple as $k = r + b \to \infty$.

The states of $J$ are now all of type $D^-$. We choose a highest weight state with $J_3 = -2r - b - n$ for Landau level $n$, expressed as

$$|-2r-b-n, -2r-b-n\rangle = \sum_{k=0}^{n} D_k\,|r, -r-n+k; r+b, -r-b-k\rangle\,. \tag{65}$$

It is a finite sum with $n + 1$ terms, always normalizable. Implementing the condition $J_+ |-2r-b-n, -2r-b-n\rangle = 0$ and the normalization condition $\sum_k |D_k|^2 = 1$, much along the lines of the subcritical case, leads to the expression

$$|D_k|^2 = \frac{n!\,\Gamma(4r+2b+n-1)\,\Gamma(2r+2b+n)\,\Gamma(2r+n)}{(n-k)!\,k!\,\Gamma(4r+2b+2n-1)\,\Gamma(2r+2b+k)\,\Gamma(2r+n-k)}\,, \tag{66}$$

and the entanglement entropy is given by the standard formula. The entropy is zero for the lowest Landau level and increases monotonically for higher Landau levels.

We summarize the important differences of the NC hyperplane from the commutative one: First, for negative $B$ (or rather $B\theta$) there is only a finite number of Landau levels that can form; and second, for $B > 1/\theta$ there are only Landau levels with no continuum and exponential divergence is completely banished.

## 4 Bosonic representation of noncommutative hyperbolic dynamics

In this section we outline bosonic representation of particle on the noncommutative hyperbolic plane. A realization of $R_i$ and $K_i$ satisfying the algebra (45) in terms of bosonic ladder operators

is given by,

$$R_+ = \sqrt{N+2r-1}\, a^\dagger, \quad R_- = a\sqrt{N+2r-1}, \quad R_3 = N+r\,, \tag{67}$$

$$K_+ = b\sqrt{M+2k-1}, \quad K_- = \sqrt{M+2k-1}\, b^\dagger, \quad K_3 = -M-k\,. \tag{68}$$

Note that the Casimirs $r$ and $k$ appear explicitly in the realization and fix the representation, which is embedded in the oscillator Fock space.

As explained in the Appendix (and the entanglement entropy section) the sign of $\gamma = 1 - \theta B$ fixes the choice of representations for $R$ and $K$. For $\gamma > 0$ (subcritical case), we choose $D_r^+$ for $R$ and $D_k^-$ for $K$, the sign $\pm$ in the superscript corresponding to different signs of $R_3$ ($K_3$). For the $\gamma < 0$ (overcritical case), we choose the representations $D_r^-$ and $D_k^-$. Even though the Hamiltonian will look different in the subcritical and the overcritical regime, there is no discontinuity as we transit from $D_r^- \otimes D_k^+$ to $D_r^- \otimes D_k^-$.

$$H_{\gamma>0} = \frac{\gamma}{2a^2}(R^2 + K^2 + R_+ K_- + R_- K_+ + 2R_3 K_3) + \frac{B^2 a^2}{2\gamma}\,. \tag{69}$$

In terms of the bosonic operators, we can write the above Hamiltonian as,

$$H_{\gamma>0} = \frac{\gamma}{2a^2}\left(\sqrt{a^\dagger a + 2r - 1}\,\sqrt{b^\dagger b + 2k - 1}\, a^\dagger b^\dagger - (a^\dagger a + r)(b^\dagger b + k) + H.c.\right)$$
$$+ \frac{\gamma}{2a^2}(r(1-r) + k(1-k)) + \frac{B^2 a^2}{2\gamma}\,. \tag{70}$$

The above Hamiltonian is non-linear in the bosonic operators. The spectrum of this complicated non-linear Hamiltonian is exactly determined by the spectrum of the operator $J^2$.

For the overcritical case, the spectrum is bounded from below if we choose $D_r^-$ for $R$ and $D_k^-$ for $K$. $D_r^-$ can be obtained by switching $R_3 \to -R_3$ and $R_+ \rightleftarrows R_-$. The Hamiltonian in terms of the bosonic operators can be expressed as,

$$H_{\gamma<0} = \frac{\gamma}{2a^2}(R^2 + K^2 + R_+ K_+ + R_- K_- - 2R_3 K_3) + \frac{B^2 a^2}{2\gamma}. \tag{71}$$

In terms of the bosonic operators,

$$H_{\gamma<0} = \frac{\gamma}{2a^2}\left(\sqrt{a^\dagger a + 2r - 1}\, a^\dagger b\,\sqrt{b^\dagger b + 2k - 1} + (a^\dagger a + r)(b^\dagger b + k) + H.c.\right)$$
$$+ \frac{\gamma}{2a^2}(r(1-r) + k(1-k)) + \frac{B^2 a^2}{2\gamma}\,. \tag{72}$$

The non-linear term inside the square root can be expanded order by order in the limit of large Casimir values $(r, k) \gg 1$. Although the above Hamiltonians contain a tower of non-linear interactions looking increasingly complicated, their spectrum can be obtained exactly from the spectrum of $J^2$ and should exhibit the full range of phenomena identified for the NC hyperplane.

# 5 Entanglement entropy transition on compact noncommutative space: NC Sphere

In the previous section, we considered the non-compact case of NC hyperplane. To make contact with the quantum chaotic dynamics in the presence of non-commutativity, we need to study quantum dynamics on a compact NC hyperplane, which will be carried out in a separate

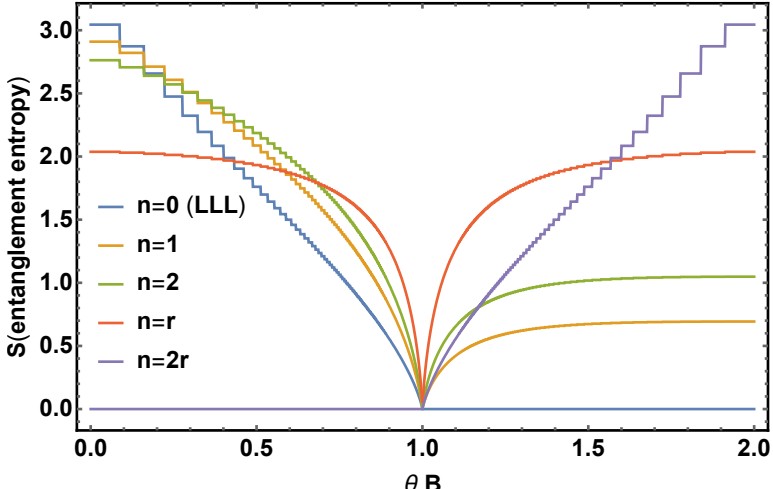

Figure 4: Entanglement entropy as a function of magnetic field for the noncommutative sphere case for LL index $n = 0, 1, 2, r, 2r$. The step like structures are due to the monopole quantization on the spherical manifold.

publication. However, to see the role of compactness on the entanglement entropy transition, we consider the simpler case of NC sphere. The NC sphere of radius $a$ in a constant magnetic field is realized in terms of two independent $SU(2)$ algebras $R_i$ and $K_i$, representing "left" and "right" action of generalized coordinates on charged noncommutative fields. $R_i$ represent the NC coordinates on the sphere and $K_i$ represent covariant coordinates that give rise to gauge fields. The angular momentum is given by

$$J_i = R_i + K_i \, , \tag{73}$$

and the Hamiltonian of a charged particle is (up to a constant) the quadratic Casimir

$$H = \frac{\gamma}{2a^2}J^2 = \frac{\gamma}{2a^2}\sum_{i=1}^{3}J_i^2 \, , \quad \gamma = 1 - \theta B = \pm\sqrt{\frac{r(r+1)}{k(k+1)}} \, . \tag{74}$$

The scaling coefficient $\gamma$ ensures recovery of the proper planar limit for $a \to \infty$. The two $SU(2)$ components $R$ and $K$ are in irreps with spins $r$ and $k$, respectively. The difference $k - r = b$ quantifies the strength of the magnetic field, and the integer $2b$ represents the magnetic monopole number, which remains quantized in the noncommutative case. The Hilbert space for a charged particle is the direct product of left and right representations, and energy eigenstates correspond to irreducible components (irreps) in this space. That is,

$$(r) \otimes (r+b) = (b) \oplus (b+1) \oplus \cdots \oplus (b+2r) \, , \tag{75}$$

where we assumed $b \geq 0$ (the case $b < 0$ can be treated similarly). The ground state corresponds to the spin-$b$ component and is $2b + 1$-degenerate – it is the LLL on the NC sphere. Note that this is an "antiferromagnetic" combination of $R$ and $K$.

We will calculate the entanglement between $R$ and $K$ for a highest weight state in the $n$th Landau level, corresponding to irreps of spin $b + n$. This state corresponds to a maximally localized state on the sphere and maximizes the value of the angular momentum along a particular direction. By rotational symmetry we may pick *any* state that has this property, and for convenience we pick the highest weight state $|b + n, b + n\rangle$ annihilated by $J_+$. In terms of product states of $R$ and $K$ it is given by the linear combination

$$|b+n, b+n\rangle = \sum_{k=-r+n}^{r} C_k \, |r, k \, ; r+b, b+n-k\rangle \, , \tag{76}$$

with $C_k$ Clebsch-Gordan coefficients (whose dependence on $r$, $b$ and $n$ has been suppressed). The two most interesting states are the lowest Landau level $n = 0$, for which the above superposition has the maximum number of terms, and the highest Landau level $n = 2r$ which is a single product state.

Applying the highest weight condition $J_+ |b + n, b + n\rangle = 0$ and using the standard $SU(2)$ expressions for the action of $R_+$ and $K_+$ on their respective states we obtain

$$C_{k+1} = -C_k \sqrt{\frac{(r-k)(r+k+1)}{(r+2b+n-k)(r-n+k+1)}} \,. \tag{77}$$

Switching to coefficients

$$D_l = C_{-r+n+l}, \quad l = 0, 1, \ldots, 2r - n \tag{78}$$

and solving the recursion relation (77) we have

$$D_l = D_0 (-1)^l \sqrt{\frac{(2r-n)!\,(n+l)!\,(2r+2b-l)!}{(2r)!\,(2b+n)!\,(2r-n-l)!\,l!}} \,. \tag{79}$$

Finally, assuming $D_0$ is real and positive and normalizing $\sum_l |D_l|^2 = 1$ gives

$$D_0^{-2} = \frac{(2r-n)!}{(2r)!\,(2b+n)!} \sum_{l=0}^{2r-n} \frac{(n+l)!\,(2r+2b-l)!}{(2r-n-l)!\,l!} = \frac{(2r+2b+n+1)!}{(2r)!\,(2b+2n+1)!} \tag{80}$$

and thus

$$D_l = (-1)^l \sqrt{\frac{(2r-n)!\,(2b+2n+1)!\,(2r-l)!\,(2b+n+l)!}{n!\,(2b+n)!\,(2r+2b+n+1)!\,l!\,(2r-n-l)!}} \,. \tag{81}$$

$|b + n, b + n\rangle$ is a pure state. Tracing over the states in $K$ leads to a mixed state for $R$ with density matrix

$$\begin{aligned}\rho &= \operatorname{tr}_K |b + n, b + n\rangle\langle b + n, b + n| \\ &= \sum_{l=0}^{2r-n} |D_l|^2 \,|r, -r + n + l, \rangle\langle r, -r + n + l| \,. \end{aligned} \tag{82}$$

The von Neuman entropy of this density matrix

$$S_n = -\sum_{l=0}^{2r-n} |D_l|^2 \ln |D_l|^2 \tag{83}$$

is the entanglement entropy between the spaces $R$ and $K$ for Landau level $n$. We obtain

$$\begin{aligned}S_n = \; &\ln \frac{(2r+2b+n+1)!\,(2b+n)!\,n!}{(2b+2n+1)!\,(2r-n)!} - \frac{(2r-n)!\,(2b+2n+1)!}{n!\,(2b+n)!\,(2r+2b+n+1)!} \times \\ &\sum_{l=0}^{2r-n} \frac{(n+l)!\,(2r+2b-l)!}{l!\,(2r-n-l)!} \ln \frac{(n+l)!\,(2r+2b-l)!}{l!\,(2r-n-l)!} \,. \end{aligned} \tag{84}$$

This is the general formula. We now focus on special cases.

*i. Sub-critical magnetic field, ground state (LLL).*

In this case the coefficient $\gamma = 1 - \theta B$ in (74) is positive and so the ground state is the lowest irrep of $J$, that is, the lowest Landau level $n = 0$ "antiferromagnetic" state in $R$ and $K$. In this case the entanglement entropy becomes

$$S_0 = \ln \frac{(2r+2b+1)!}{(2b+1)(2r)!} - \frac{(2b+1)(2r)!}{(2r+2b+1)!} \sum_{l=0}^{2r} \frac{(l+2b)!}{l!} \ln \frac{(l+2b)!}{l!} \,. \tag{85}$$

A few comments are in order:

a) For $b = 0$ (zero magnetic field) we recover $S = \ln(2r + 1)$, the log of the number of area cells on the NC sphere.

b) The entanglement entropy decreases monotonically with the magnetic field strength.

c) In the limit $r \gg 1$, $k \gg 1$, $r/k = \gamma$ we recover the result for the entanglement entropy on the NC plane. The easiest way to see it is to return to the formula (81) for $D_l$ and take the large-$r$, large-$b$, $r/(r + b) \to \gamma$ limit. We obtain

$$|D_l|^2 = \left( \frac{b}{r+b} \right)^{n+1} \left( \frac{r}{r+b} \right)^l \frac{(n+l)!}{n!\, l!} \tag{86}$$

and, putting $b/r = 1 - \gamma$ and $r/(r + b) = \gamma$, $|D_l|^2$ become identical to the planar coefficients (34) upon putting $\tanh\lambda = \gamma$ and $l = k$.

    *ii.* Sub-critical magnetic field, higher excited states.

    This corresponds to higher Landau levels $n > 0$ and the full formula (84) applies. Important properties to note is that the entanglement initially still decreases monotonically with the magnetic field. However, it initially increases with the Landau level $n$, reaches a maximum and then monotonically decreases.

    *iii.* Overcritical magnetic field

    The entanglement entropy changes crucially at the critical point $\theta B = 1$, that is, $\gamma = 0$. On the NC sphere taking taking $\gamma \to 0$ means $r/k \to 0$, that is, $k \to \infty$ or $N \to \infty$. To see what happens to $S$, we note that in this limit the ($2r + 1$ in number) Clebsch-Gordan coefficients become

$$D_l \to 0 \;\; (l > 0)\,, \quad D_0 \to 1\,. \tag{87}$$

Therefore, in that limit the density matrix $\rho$ becomes a pure state and $S_n \to 0$. The entanglement entropy for all Landau levels vanishes.

    Overcritical magnetic fields, $\theta B > 1$, correspond to finite $b$. However, since the prefactor $\gamma = 1 - \theta B$ becomes negative, the Hamiltonian is *inverted* (see [16, 17] for details). The irreps of $R$ and $K$ are the same but now correspond to $\gamma = -\sqrt{r(r+1)/k(k+1)}$, that is, to a magnetic field $B' = 2/\theta - B$. The ground state is now the *maximal* representation of $J$, $2r + b$, that is, $n = 2r$. This is a "ferromagnetic" state, and it consists of $4r + 2b + 1$ states. Its highest weight state is

$$|b + 2r, b + 2r\rangle = |r, r\,; r + b, r + b\rangle\,. \tag{88}$$

This is a product state and the entanglement entropy is zero.

    Excited states correspond to *lower* values of $n = 2r - 1, 2r - 2, \ldots$. The entanglement entropy of an excited state $n$ is the same as that for the Landau level $2r - n$ in the subcritical case for the dual magnetic field $B' = 2/\theta - B$. We plot entanglement entropy transition for $n = 0, 1, 2, r, 2r$ in Fig. 4.

# 6 Conclusions and outlook

In this work, we computed Lyapunov exponents and the entanglement entropy across a spectral transition on the non compact noncommutative hyperbolic plane in the presence of magnetic field. The entanglement entropy is computed between the 'left' and 'right' copies of the spatial degrees of freedom.

    One of the main motivations of this work was to explore possible deformations of the Schwarzian theory, which emerges in the low energy sector of the Sachdev-Ye-Kitaev model. The Schwarzian action is closely related to the Landau level problem on the hyperbolic plane. The LL problem on the hyperbolic plane has two competing effects; namely, the negative

curvature-induced exponential divergence of operator growth (trajectories in the semiclassical sense) and the magnetic field-induced bounded operator growth (closed cyclotron orbits in the semiclassical sense). The energy spectrum consists of a continuous part associated with the exponentially diverging states and a discrete part corresponding to cyclotron orbits. The Schwarzian limit of the LL problem consists only of the continuous part of the spectrum and is realized by an appropriate coordinate rescaling followed by taking the large magnetic field limit, as shown in Ref. [11]. An interesting question would be to interpolate between the LL problem on the noncommutative hyperbolic plane and an appropriate Schwarzian theory, while retaining the spectral transition. We further showed that using bosonic representation of SL(2,R) algebra, we can construct solvable non-linear bosonic Hamiltonians corresponding to the Landau level problem on the NC hyperplane. The curvature, noncommutativity and the magnetic field are encoded as parameters of the solvable bosonic Hamiltonian.

Our work is also a step towards quantifying chaos on the noncommutative pseudosphere (compact hyperbolic plane). In order for the exponential divergence with positive Lyapunov exponents to be an indicator of quantum chaos one needs to invoke phase space mixing by compactifying the noncommutative space to a genus-$g$ noncommutative manifold through reducing by a specific discrete group [15]. Such a compactification is also necessary to make a more direct connection to Schwarzian-like theories. As a precursor to this effort we explored the dynamics of a particle on the compact noncommutative sphere, and for completeness we also treated the noncommutative plane. We showed how the spectral transition manifests on the noncommutative sphere and how the entanglement entropy behaves across this transition. In addition to studying chaotic dynamics on the compact non-commutative genus-$g$ manifold, it would be interesting to see if such dynamics can be realized as a low energy limit of SYK-like quantum many body models. We leave the treatment of compact noncommutative hyperbolic spaces and related questions for future work.

# Acknowledgements

We would like to thank Victor Galitski, Efim Rozenbaum, David Huse and Herman Verlinde for interesting discussions. The research of S.G. was supported by NSF OMA-1936351 and a PSC CUNY grant. A.P. acknowledges the support of NSF under grant 1519449. S.G. also acknowledges the hospitality of the Aspen Center for Physics (ACP) where part of this work was carried out; ACP is supported in part by NSF grant PHY-1607611.

# 7   Appendix: Review of NC quantum mechanics

Noncommutative spaces are realized in terms of non-commuting coordinates and their corresponding derivatives. The simplest example is the NC plane, with two space coordinates satisfying

$$[x_1, x_2] = i\theta \,, \tag{89}$$

with $\theta$ a (commuting) noncommutativity parameter. We assume that $x_1, x_2$ act on a single irreducible representation of the Heisenberg algebra (multiple copies would correspond to several overlapping NC planes and would lead to a nonabelian structure).

Functions on the NC plane $\Phi$ become operators on the Hilbert space and coordinates act on them through left multiplication. Derivatives are realized through their action in the Heisenberg picture, that is, as commutators

$$\partial_1 \Phi = \frac{i}{\theta}[x_2, \Phi]\,, \quad \partial_2 \Phi = -\frac{i}{\theta}[x_1, \Phi] \tag{90}$$

that act as ordinary derivatives on any monomial of $x_1, x_2$.

We can trade the appearance of commutators for a doubling of the space coordinates by considering the right action of $x_1, x_2$ on functions as independent operators; that is,

$$\Phi x_1 = -k_1 \Phi \,, \quad \Phi x_2 = -k_2 \Phi \,. \tag{91}$$

Under this definition $x_i$ and $k_j$ commute while $k_1, k_2$ constitute another NC plane:

$$[x_1, x_2] = -[k_1, k_2] = i\theta \,, \quad [x_i, k_j] = 0 \,. \tag{92}$$

Derivatives are expressed in terms of $x_i$ and $k_i$ as

$$\partial_1 = \frac{i}{\theta}(x_2 + k_2) \,, \quad \partial_2 = -\frac{i}{\theta}(x_1 + k_1) \,, \quad [\partial_i, x_j] = \delta_{ij} \,, \quad [\partial_i, \partial_j] = 0 \,. \tag{93}$$

We refer to $x_i$ and $k_i$ as "left" and "right" coordinates. They each act on their own Hilbert space, and functions $\Phi$ can be though of as states on the direct product of the two spaces via the mapping

$$\Phi = \sum_{m,n} \Phi_{mn} |m\rangle \langle n| \quad \rightarrow \quad |\Phi\rangle = \sum_{m,n} \Phi_{mn} |m\rangle \otimes |n\rangle \,, \tag{94}$$

with $|n\rangle$ a complete set of states on the Heisenberg Hilbert space.

Quantum mechanically the momenta of a particle on the NC plane are represented as usual by $p_j = -i\partial_j$ and are expressed as

$$p_1 = \frac{1}{\theta}(x_2 + k_2) \,, \quad p_2 = -\frac{1}{\theta}(x_1 + k_1) \,, \quad [x_i, p_j] = i\delta_{ij} \,, \quad [p_i, p_j] = 0 \,. \tag{95}$$

The wavefunction of the particle becomes a function on the NC plane and is an element of the product of left and right Hilbert spaces. (The same result is reached by simply looking for an irreducible representation of the NC coordinates $x_i$ and the momenta $p_i$ satisfying the commutation relations (95).) We stress that the two Hilbert spaces do not represent a doubling of degrees of freedom: the Hilbert spaces of the two commutative coordinates and their derivatives $x_1, p_1$ and $x_2, p_2$ have been traded for the left and right Hilbert spaces.

A constant magnetic field can be introduced by modifying the momentum algebra as in the commutative case

$$[p_i, p_j] = iB \,. \tag{96}$$

This can be achieved by modifying the expressions of $p_i$ in terms of $x_i$ and $k_i$

$$p_1 = \frac{1}{\theta}(x_2 + \sqrt{\gamma}\,k_2) \,, \quad p_2 = -\frac{1}{\theta}(x_1 + \sqrt{\gamma}\,k_1) \,, \quad \gamma = 1 - \theta B \,. \tag{97}$$

NC manifolds with constant curvature, namely NC sphere and NC hyperbolic plane, are constructed in an analogous way. Their commutative construction involves coordinates $X_i$ satisfying the constraint

$$X_1^2 + X_2^2 \pm X_3^2 = \epsilon\, a^2 \tag{98}$$

in an ambient space with metric

$$ds^2 = dX_1^2 + dX_2^2 + \epsilon\, dX_3^2 \,, \tag{99}$$

with $\epsilon = +1$ ($-1$) corresponding to a spherical (hyperbolic) space respectively. NC coordinates satisfy (98) and commute to the corresponding isometry group

$$[X_i, X_j] = i\,\frac{\theta}{a}\,\epsilon_{ijk} X^k \,, \tag{100}$$

such that, when $X_3 \simeq a \gg X_1, X_2$ it reproduces the planar NC relation $[X_1, X_2] = i\theta$. This means that $R_i = (a/\theta)X_i$ satisfies the corresponding $SU(2)$ or $SL(2,R)$ algebra with Casimir

$$[R_i, R_j] = i\epsilon_{ijk}R^k , \quad R_i R^i = r(\epsilon r + 1) = \epsilon \frac{a^4}{\theta^2} . \tag{101}$$

So the curvature of space $\pm 1/a^2$ and noncommutativity parameter $\theta$ are encoded in the Casimir or $R_i$. Functions on the NC spaces are represented by operators acting on the irrep of $R_i$ with the above Casimir. In analogy with the NC plane we define a new set of $SU(2)$ or $SL(2,R)$ generators $K_i$ that represent the right action of $R_i$ on functions and obey the same commutation relations:

$$\Phi R_i = -K_i \Phi , \quad [K_i, K_j] = i\epsilon_{ijk}K^k \tag{102}$$

(Note that the minus sign in the definition of $K_i$ is crucial for it to satisfy the same algebra as $R_i$, since it represents right action.) Momentum operators become the generators of the global rotations or $SL(2,R)$ transformations and are given by

$$J_i = R_i + K_i , \quad [J_i, R_j] = i\epsilon_{ijk}R^k , \quad [J_i, J_j] = i\epsilon_{ijk}J^k . \tag{103}$$

This should be compared with (95) for the planar case.

An important difference arises when incorporating magnetic fields. Unlike the plane, this does not involve changing the algebra of momenta but rather changing the Casimirs (this is true also in the commutative case). A semi-heuristic way to identify the necessary modification is to consider the well-known relation between angular momentum and coordinates in the presence of a constant magnetic field $B$

$$X_i J^i = -\epsilon B a^3 . \tag{104}$$

This holds for a sphere of radius $a$ and remains true for the hyperbolic plane. In the NC case this translates to

$$\tfrac{1}{2}(X_i J^i + J^i X_i) = -\epsilon B a^3 \tag{105}$$

the symmetrization being necessary to deal with the NC nature of the fields and render the product Hermitian. In terms of $R_i = (a/\theta)X_i \cdot$ and $K_i$ the above relation becomes

$$\tfrac{1}{2}R_i(R^i + K^i) - \tfrac{1}{2}(R^i + K^i)K_i = \tfrac{1}{2}(R^2 - K^2) = -\epsilon \frac{Ba^4}{\theta} . \tag{106}$$

This means that the Casimirs $R^2 = r(\epsilon r + 1)$ and $K^2 = k(\epsilon k + 1)$ cannot be equal any more but must satisfy

$$(k - r)(k + r + \epsilon) = \frac{2Ba^4}{\theta} . \tag{107}$$

In the commutative limit $r, k \gg 1$ while $k - r = b$ remains of order 1, and $r \simeq a^2/\theta$, so

$$b = k - r = Ba^2 . \tag{108}$$

On the sphere the above implies

$$2b = 2Ba^2 = \frac{B\, 4\pi a^2}{2\pi} = M . \tag{109}$$

So $2b$ is the monopole number $M$, and since $b$ for $SU(2)$ is a half-integer we obtain the monopole quantization condition, which remains valid in the NC case. For the hyperplane, on the other hand, $r$ and $k$ need not be quantized and we have no monopole quantization, as expected on an open infinite space.

The Hamiltonian is the kinetic energy of the particle and can be obtained by considering the corresponding NC gauge theory on the sphere or hyperplane (for details we refer the reader to [15–18]):

$$H = \frac{\gamma}{2a^2}\left[J^2 - \epsilon\left(\frac{Ba^2}{\gamma}\right)^2\right], \quad \text{with} \quad \gamma = 1 - \theta B = \pm\sqrt{\frac{r(r+\epsilon)}{k(k+\epsilon)}} \tag{110}$$

or, more explicitly in terms of the Casimirs $r$ and $k$,

$$H = \pm\frac{1}{2a^2}\sqrt{\frac{r(r+\epsilon)}{k(k+\epsilon)}}\left(J^2 - \epsilon\left[\sqrt{k(k+\epsilon)} \mp \sqrt{r(r+\epsilon)}\right]^2\right). \tag{111}$$

Apart from factors and a constant shift it is simply the Casimir $J^2$ and can be obtained by pure group theory by decomposing $J = R + K$ into irreps of $SU(2)$ or $SL(2,R)$.

An important point is that the same two Casimirs $r, k$ for $R$ and $K$ describe two different magnetic fields, corresponding to the $\pm$ signs in (110,111), and the coefficient $\gamma$ becomes negative for $\theta B > 1$. This requires a different treatment for each of the two manifolds.

For the sphere, $R$ and $K$ are in the (unique) $SU(2)$ irreps with spins $r$ and $k$ and $J^2$ has a unique decomposition into $SU(2)$ irreps. This means that the energy spectrum for $B' = -B + 2/\theta$ is the reverse of that for $B$ and shifted upwards by a constant.

For the NC hyperplane the situation is subtler: for each value of the quadratic Casimir of $SL(2,R)$ there are two irreps, denoted $D^\pm$, depending on the sign of $R_3$ or $K_3$. For $\gamma > 0$, correspondence with the commutative case requires choosing irreps $D_r^+ \otimes D_k^-$ (the opposite choice is also possible and equivalent) and the spectrum is positive definite and unbounded from above, comprising a set of discrete Landau levels as well as a continuum. The transition to negative $\gamma$ requires the choice $D_r^- \otimes D_k^-$ in order to have a positive definite spectrum and to recover the correct commutative limit. In that case the spectrum is again unbounded from above and consists entirely of discrete Landau levels.

We conclude with the remark that an alternative formulation of NC spaces is in terms of star-products, a noncommutative operation between ordinary functions that encodes the nontrivial space commutation relations. Star products, however, become more involved on spaces of nonzero curvature, obscure the inherent simplicity of the operator formulation and miss the obvious group structure of the states. We consider these too steep a price for the comfort of working in the more familiar setting of commutative functions and restrict our exposition to the operator formulation.

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
