# Peer review of "Lyapunov exponents and entanglement entropy transition on the noncommutative hyperbolic plane"

_SciPost Physics, doi:SciPost Phys. Core 3, 003 (2020)_

## Round 2 · Referee Report · Anonymous (Referee 1) · 2020-5-20

Report

In this article, the Authors explore a connection between Lyapunov exponent and entanglement entropy in a dynamical system defined on a non-commutative hyperbolic geometry, with a magnetic field. They exhibit an explicit connection between the behaviour of entanglement entropy in the lowest Landau level and the transition between exponentially divergent and oscillatory trajectories. This is an interesting aspect that should be published. However, I would like the Authors address the following comments in a revised version:

  1. By trajectories, I believe the Authors simply mean the expectation value of certain operators. It may be useful to clarify this for better readability.

  2. Is there any way to physically understand the results discussed below equation (25)? Specifically, e.g. why is there a divergent piece in the entanglement entropy at zero magnetic field?

  3. The duality, presented in eqn (39) appears mysterious. What is the underlying reason behind this? Comments addressing this will also improve the readability of the paper.

  4. There is a generic point that I have not quite understood. In the presence of a magnetic field, specially in the large magnetic field limit, a spatial non-commutativity appears. This is different from the spatial non-commutativity described in this article. It will be very helpful if the Authors could clarify the differences/similarities between the physics of these two non-commutative aspects.

  • validity: -
  • significance: -
  • originality: -
  • clarity: -
  • formatting: -
  • grammar: -

Author:  Sriram Ganeshan  on 2020-07-17  [id 890]

(in reply to Report 1 on 2020-05-20)

**The referee writes:** In this article, the Authors explore a connection between Lyapunov exponent and entanglement entropy in a dynamical system defined on a non-commutative hyperbolic geometry, with a magnetic field. They exhibit an explicit connection between the behaviour of entanglement entropy in the lowest Landau level and the transition between exponentially divergent and oscillatory trajectories. This is an interesting aspect that should be published. However, I would like the Authors address the following comments in a revised version:

**Our response:** We thank the referee for carefully reviewing the manuscript and recommending it for publication. We also thank the referee for comments that have helped us improve the manuscript. Below are the responses to the specific comments.

**The referee writes:** By trajectories, I believe the Authors simply mean the expectation value of certain operators. It may be useful to clarify this for better readability.

**Our response:** The referee is right to point out that the trajectories are an appropriate terminology for classical systems. The Lyapunov exponent in the classical systems is extracted from the exponential divergence of trajectories. For the quantum case, we extract the Lyapunov exponent by computing the rate of operator growth, which upon taking the expectation value is closely related to the classical Lyapunov exponent. We have clarified the distinction between the classical and quantum Lyapunov diagnostics in the revised text.

**The referee writes:** Is there any way to physically understand the results discussed below equation (25)? Specifically, e.g. why is there a divergent piece in the entanglement entropy at zero magnetic field?

**Our response:** To gain some intuition into the entanglement entropy relation, consider the non-commutative parameter $\theta$ to be small, that is $(B\theta, \omega \theta)\ll 1 $. Then the expansion in $\theta$ of Eq.~18 in the paper's text gives,
\begin{align}
\tanh2\lambda=1-\frac{B^2\theta^2}{8}+\frac{\omega^2\theta^2}{2}
\end{align}
In the $\lambda \gg 1$ limit, we have
\begin{align}
e^{2\lambda}=\frac{\theta}{2}\sqrt{\frac{B^2}{4}+\omega^2}
\end{align}
In this limit, the entanglement entropy in Eq.~(25) is given by
\begin{align}
S\approxeq 1-\ln\left[2\theta \sqrt{\frac{B^2}{4}+\omega^2}
\right]
\end{align}
$\omega_B=\sqrt{\frac{B^2}{4}+\omega^2}$ is the inverse square of the spatial extent of the ground state wave function ($e^{-\frac{1}{2}\omega_B r^2}$) and $\theta$ is the fundamental quantum/cell of area on the noncommutative space. So, $N=\pi R^2/\theta\sim \frac{1}{\theta \omega_B}$ is the number of area cells spanned by the ground state wavefunction. This number of spanned area cells determines the entanglement entropy $S=\ln N$ up to an overall constant. (For finite values of $\theta$ the formula for S includes various corrections.) For $\omega=0, B\rightarrow 0$, the magnetic length is infinite and the ground state spans divergently large number of area cells leading to $S\rightarrow \infty$. For fixed $\omega$ the maximum entropy is for the case $B=0$. For large $|B|$, the magnetic length shrinks and the entropy decreases with the decreasing number of area cells spanned by the ground state wave function. But for $B\rightarrow 1/\theta$ strong non commutative effects take over and results in $S\rightarrow 0$. The effective ``commutative" (Seiberg-Witten mapped) magnetic field is $\tilde B=B/(1-\theta B) \rightarrow \infty$ as $B\rightarrow 1/\theta$ which implies $S\rightarrow 0$.

**The referee writes:** The duality, presented in eqn (39) appears mysterious. What is the underlying reason behind this? Comments addressing this will also improve the readability of the paper.

**Our response:** The duality $S(\gamma)=s(\gamma^{-1})$ for $\omega=0$ can be understood as a mapping between the ``left" and ``right" Hilbert spaces with respect to which the entanglement entropy is calculated.

From Eq.~(13) in the main text, consider the momenta $p_{1,2}$ with $\gamma>0$,
\begin{align}
p_1=\frac{1}{\sqrt{\theta}}\left(b_1+\sqrt{\gamma}\alpha_2\right)\\
p_2=\frac{1}{\sqrt{\theta}}\left(-\alpha_1-\sqrt{\gamma}b_2\right)
\end{align}
We now define new operators $(\tilde \alpha_1, \tilde b_1)=(b_1,-\alpha_1)$ and $(\tilde \alpha_2, \tilde b_2)=(-b_2,\alpha_2)$, in terms of which the momenta $p_{1,2}$ can be written as,
\begin{align}
\frac{1}{\sqrt{\gamma}} p_1=\frac{1}{\sqrt{\theta}}\left(\tilde b_2+\frac{1}{\sqrt{\gamma}}\tilde\alpha_1\right)\\
-\frac{1}{\sqrt{\gamma}} p_2=\frac{1}{\sqrt{\theta}}\left(-\tilde\alpha_2-\frac{1}{\sqrt{\gamma}} \tilde b_1\right)
\end{align}
Thus a rescaling and a parity transform $p_2\rightarrow -p_2$ of the momentum, and a mapping of spaces $1\leftrightarrow 2$ (with some canonical transformation of coordinates in each) amount to $\gamma\rightarrow 1/\gamma$. Since the Hamiltonian $H=\frac{1}{2}(p_1^2+p_2^2)$ maps to $H/\gamma $ under the above transformation, the ground state remains unchanged and has the same entanglement entropy. Note that the coordinates $x_{1,2}$ map to themselves under the same transformation. Thus for $\omega \neq 0$ the Hamiltonian $H=\frac{1}{2}(p_1^2+p_2^2)+\frac{\omega^2}{2}(x_1^2+x_2^2)$ does not map to a scaled version of itself and thus breaks the duality. Another way to express the duality is by using the ``Seiberg-Witten" mapped magnetic field $\tilde B=B/(1-\theta B)$, which leads to $S(B)=S(-\tilde B)$.

**The referee writes:** There is a generic point that I have not quite understood. In the presence of a magnetic field, specially in the large magnetic field limit, a spatial non-commutativity appears. This is different from the spatial non-commutativity described in this article. It will be very helpful if the Authors could clarify the differences/similarities between the physics of these two non-commutative aspects.

**Our response:** Non commutativity in the lowest Landau level arises from the projection operation into the degenerate ground state manifold and consequently the coordinates become equivalent to the momentum. This projection makes the momentum non-dynamical.

In NC spaces, we have {\it independent} coordinates and momenta with a deformed commutation relation between the coordinates via the non-commutativity parameter $\theta$ as well as the usual non-commutativity of the momenta via the magnetic field $B$. Thus we have {\it two} length scales in the problem, $1/\sqrt{B}$ and $\sqrt{\theta}$, which results in the interesting properties discussed in this paper, with the special point of $B=1/\theta$ where the two length scales become equal.

**List of changes**
Modified the terminology of trajectory throughout the paper with operator growth in quantum dynamics.\\
Added discussion clarifying the discussion comparing the NC physics discussed in this work with the NC arising in the Lowest Landau Level physics after projection.\\
Added footnote elaborating on the duality presented in Eqn (39).

---

## Round 3 · Referee Report · Anonymous · 2020-7-31

Report

The Authors have addressed my points and comments provided in the earlier report, and at this point I have no further comments to add.

Given this, I would recommend this updated manuscript to be published accordingly, since it easily meets the criteria for publication in this journal.

---

## Editorial Decision

published